# A comprehensive, improved protocol for generating common bean (*Phaseolus vulgaris* L.) transgenic hairy roots and their use in reverse-genetics studies

**Ronal Pacheco[1], Georgina Estrada-Navarrete[1], Jorge Solis-Miranda[1¤], Noreide Nava[1], MA Juárez-Verdayes[2], Yolanda Ortega-Ortega[3], Carmen Quinto[1] ***

**1** Departamento de Biología Molecular de Plantas, Instituto de Biotecnología, Universidad Nacional Autónoma de México, Cuernavaca, Morelos, México, **2** Departamento de Ciencias Básica, Universidad Autónoma Agraria Antonio Narro, Saltillo, Coahuila, México, **3** Departamento de Biociencias y Agrotecnología, Centro de Investigación Química Aplicada, Saltillo, Coahuila, México

¤ Current address: Instituto de Bioquímica Vegetal y Fotosíntesis, Consejo Superior de Investigaciones Científicas (CSIC)-Universidad de Sevilla, Sevilla, España

* carmen.quinto@ibt.unam.mx

## Abstract

Generating transgenic hairy roots has been the preferred strategy for molecular studies in common bean (*Phaseolus vulgaris* L.), since generating stable knockout lines in this species is challenging. However, the number of plants producing hairy roots following the original protocol published in 2007 is usually low, which has impeded progress. Since its initial publication, the original protocol has been extensively modified, but these modifications have not been adequately or systematically reported, making it difficult to assess the reproducibility of the method. The protocol presented here is an update and expansion of the original method. Importantly, it includes new, critical steps for generating transgenic hairy roots and using them in molecular analyses based on reverse-genetics approaches. Using this protocol, the expression of two different genes, used as an example, was significantly increased or decreased in approximately 30% of the transformed plants. In addition, the promoter activity of a given gene was observed, and the infection process of rhizobia in transgenic hairy roots was monitored successfully. Thus, this improved protocol can be used to upregulate, downregulate, and perform promoter activity analysis of various genes in common bean transgenic hairy roots as well as to track rhizobia infection.

## Introduction

The ability of legume species (Fabaceae) to establish a symbiosis with mycorrhizal fungi and rhizobia through their roots allows them to thrive in nutrient-poor soils. In addition, some legume species, such as common bean (*Phaseolus vulgaris* L.), are important sources of protein and are used worldwide for human consumption. Common bean can establish symbiotic relationships with arbuscular mycorrhizal fungi and nitrogen-fixing bacteria commonly called

**Data Availability Statement:** All relevant data are within the paper and its Supporting Information files.

**Funding:** R. P: Consejo Nacional de Humanidades Ciencias y Tecnologías (CONAHCyT) 0457178158. https://conahcyt.mx/ C. Q: Programa de Apoyo a Proyectos de Investigación e Innovación Tecnológica, Universidad Nacional Autónoma de México (PAPIIT/UNAM) IN203021. https://dgapa. unam.mx/index.php/impulso-a-la-investigacion/ papiit The funders did not and will not have a role in study design, data collection and analysis, decision to publish, or preparation of the manuscript.

**Competing interests:** The authors have declared that no competing interests exist.

rhizobia [1]. This crop species also has agronomic and nutritional values, making it a widely used experimental model to study the mechanisms that regulate these symbiotic interactions. In particular, to better understand the molecular basis of bean–mycorrhiza and bean–rhizobia symbioses, two reverse-genetics approaches are commonly used, gene overexpression and gene silencing, together with promoter activity analysis using reporter genes [2–5]. These methods require the generation of transgenic hairy roots in wild-type plants (hereafter referred to as composite plants) via *Agrobacterium rhizogenes*–mediated transformation.

A viable protocol for generating composite common bean plants was published in 2007 [6], and the original protocol has been optimized over the years. We published a compendium of all updated steps for this protocol as supplementary material in a recent article [7]. However, even this updated material lacked detailed information on how to successfully and reliably generate transgenic hairy roots. This work presents an improved, updated, expanded, and detailed protocol for the generation of transgenic hairy roots in common bean. This protocol includes improvements made over the previous one, with the aim of making it simpler and reproducible. To do this, we used constructions of plasmids previously obtained by our group as an example of the improvements generated with this modified protocol. Furthermore, we discuss critical steps for applying this protocol in reverse-genetics-based experiments to study the molecular mechanisms regulating mycorrhizal and rhizobial symbioses in common bean.

Understanding the molecular mechanisms underlying the establishment of these symbiotic interactions is crucial for improving sustainable agriculture practices. In this sense, reverse-genetics-based approaches represent useful tools to functionally characterize genes of interest during symbiotic interactions.

## Materials and methods

The protocol described in this peer-reviewed article is published on protocols.io, **https://dx. doi.org/10.17504/protocols.io.261ge3bpjl47/v2**, and is included for printing as S1 File with this article.

### Plant growth conditions and inoculation with rhizobia

Composite plants were generated as described in the protocol published at protocols.io, **https://dx.doi.org/10.17504/protocols.io.261ge3bpjl47/v2.** The plasmid constructs used in this protocol were previously constructed by our group. We used them here for convenience to illustrate how transgenic roots are obtained using methodologically proven plasmid constructs. To quantify the transcript levels of a given gene, for example, the patatin-related phospholipase A gene *pPLA-IIγ* (Phvul.001G020300.1) in transgenic hairy roots overexpressing this gene, composite plants were inoculated with *Rhizobium tropici* CIAT 899 and transgenic hairy roots were harvested at 7 days post inoculation (dpi). To quantify the transcript levels of a second gene, for example, the phospholipase D gene *PvPLDα2* (Phvul.005G177300.1), we used 18-day-old transgenic hairy roots that downregulate this gene without inoculation with the bacteria.

To determine the promoter activity of a given gene, such as the RPM1-induced protein kinase gene *PvRIPK2* (Phvul.003G130400.1), it was analyzed at 7 dpi with *R. tropici* transformed with the *β-glucuronidase* gene (GUS). The analysis of the advancement of the infection thread (IT) in the cortical region of transgenic hairy roots was performed transgenic hairy roots overexpressing *PvAtg6* (Phvul.005G029900.1), an autophagy-related gene. Transgenic hairy roots overexpressing PvAtg6 were inoculated with *R. tropici* CIAT899 expressing red fluorescent protein (Ds-red) and observed using a confocal microscope at 8 dpi.

## RNA extraction and reverse-transcription quantitative PCR (RT-qPCR) analysis

RNA extraction and complementary DNA synthesis were performed using the protocol published on protocols.io, **https://dx.doi.org/10.17504/protocols.io.8epv5jq24l1b/v1**. qPCR assays were performed using Maxima SYBR Green/ROX qPCR Master Mix (2X) (Thermo Scientific, Waltham, MA, USA) and a qPCR system (QuantStudio 5; Applied Biosystems, Waltham, MA, USA). The qPCR steps were as follows: 95˚C for 10 min and 30 cycles of 95˚C for 15 s and 60˚C for 60 s. The relative transcript levels of each gene were calculated using the elongation factor 1α gene (*EF1α*, Phvul.004G075100.1) as a reference and the $2^{-\Delta\Delta C_T}$ method [8].

## Plasmid construction

The plasmid used to analyze the *PvRIPK*2 promoter was generated by inserting the 2-kb promoter region of *PvRIPK*2 upstream of the initiation codon into the cloning vector pENTR/D-TOPO (Invitrogen, Life Technologies, Carlsbad, CA, US); subsequently, this insert was transferred into the target vector pBGWSF7.0, which contains GUS in-frame with the inserted region [9], by homologous recombination. To generate the plasmids used to overexpress *pPLA-IIγ* (35S:*pPLA-IIγ-EGFP*) and the autophagy-related gene in common bean (35S:*PvAtg6-EGFP*), the respective coding sequence (CDS) was inserted into pENTR/D-TOPO. Each resulting plasmid was recombined with the destination vector pH7FWG2D [9]. The resulting plasmid contained an open reading frame driven by the 35S promoter and the respective CDS in-frame and upstream of the enhanced green fluorescent protein gene (*EGFP*). To downregulate *PvPLDα2* (encoding phospholipase D), a 146-bp fragment of this gene was inserted from the 3' untranslated region in pENTR/D-TOPO and recombined in an inverted repeat orientation into the binary vector pTDT-DC-RNAi [10], resulting in the *PvPLDα2*-RNAi plasmid.

To express the *Cas9* gene in common bean transgenic hairy roots, the CDS of *Cas9* was inserted into pENTR-TOPO. The *Cas9* CDS and the Gateway cloning system cassette were transferred into pH7FWG2 to build an expression vector for Gateway cloning suitable for *Agrobacterium rhizogenes*–or *A. tumefaciens*–mediated plant transformation. In all cases, Gateway technology LR Clonase II was used [9]. All constructs have been used to define the function of the corresponding genes during rhizobial symbiosis and have not yet been published.

## Expected results

Using this protocol, transgenic hairy roots can be generated in common bean in approximately 16 days. In our laboratory, we have successfully used this method to obtain hairy roots using *A. rhizogenes* strains transformed with specific plasmids for overexpression and RNA interference (RNAi)–mediated gene silencing [5, 7]. Fig 1 shows the transcript levels of *pPLA-IIγ* and *PvPLDα2* in common bean transgenic hairy roots transformed with the 35S:*pPLA-IIγ-EGFP* or *PvPLDα2*-RNAi plasmid, respectively. The transcript level of the *pPLA-IIγ* gene was significantly higher in plants that overexpress this gene (35S:*pPLA-IIγ-EGFP*) and significantly lower in plants with RNAi of *PvPLDα2* (*PvPLDα2*-RNAi) compared to the control. These results indicate that hairy root transformation is a viable strategy to overexpress and downregulate genes of interest in various studies using reverse genetics to investigate several processes, including legume–rhizobia symbiosis.

As an example of the use of the protocol described here, we analyzed the promoter activity of the *PvRIPK2* gene and followed the progress of the IT through which rhizobia migrate toward the cortical zone in transgenic hairy roots that overexpress the *PvAtg6* gene. As shown

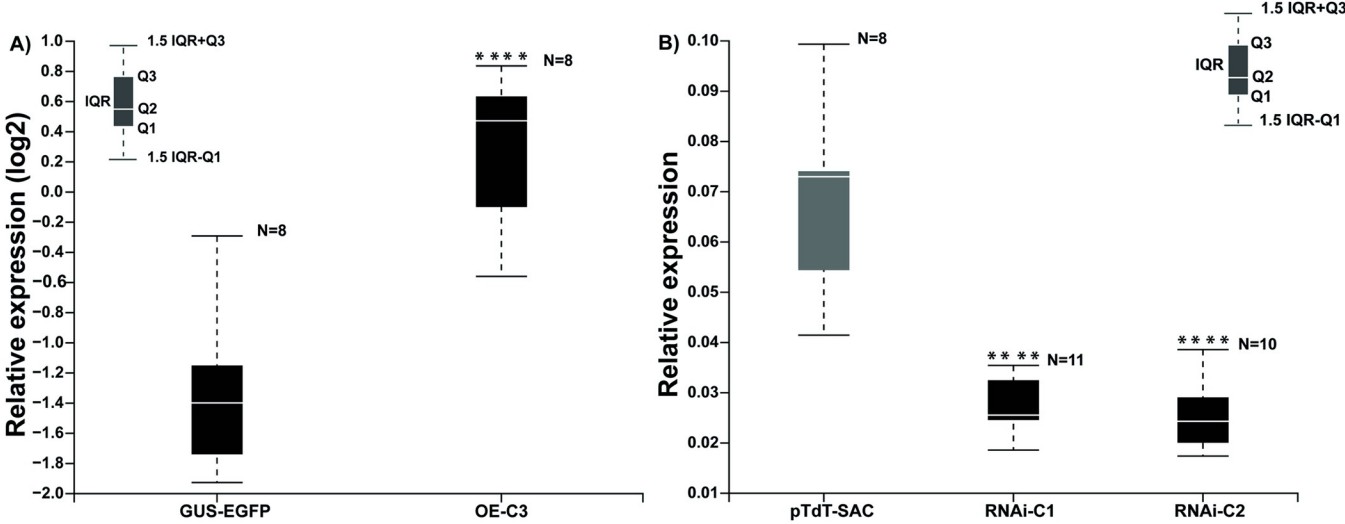

**Fig 1. Quantification of the transcript abundance of two genes encoding common bean phospholipases in transgenic hairy roots.** A) Relative expression levels of *pPLA-IIγ* in transgenic hairy roots carrying the overexpression plasmid 35S:*pPLA-IIγ-EGFP* (OE-C3) or a control plasmid (GUS-EGFP). B) Relative expression levels of *PvPLDα2* in transgenic hairy roots carrying *PvPLDα2*-RNAi, an RNAi-based plasmid for *PvPLDα2* (RNAi-C1 and RNAi-C2 lines), or a control plasmid (pTdT-SAC). The elongation factor gene *EF1α* was used as an endogenous reference gene. Q, quartile; IQR, interquartile range. For statistical analysis, a Montecarlo simulation test was used with 9,999 resamples without replacement (**** $p \leq 0.0001$). *N* indicates the sample size of three biological replicates.

in Fig 2, *PvRIPK2* promoter activity was observed in the rhizobia infection zone and in nodule primordia (Fig 2), indicating that GUS activity can be observed in different cell layers of transgenic hairy roots. Furthermore, we clearly observed the progress of ITs by examining the emission of fluorescence in epidermal cells (Fig 3). These results demonstrate the usefulness of transgenic hairy roots, in this case, to study rhizobial symbiosis using histochemical and microscopic methods. Notably, it was possible to generate transgenic hairy roots expressing the clustered regularly interspaced short palindromic repeats (CRISPR)-associated nuclease 9 gene *Cas9* (Fig 4). Taken together, these results indicate that our updated protocol can be used to functionally characterize genes of interest in transgenic hairy roots during legume–rhizobia symbiosis.

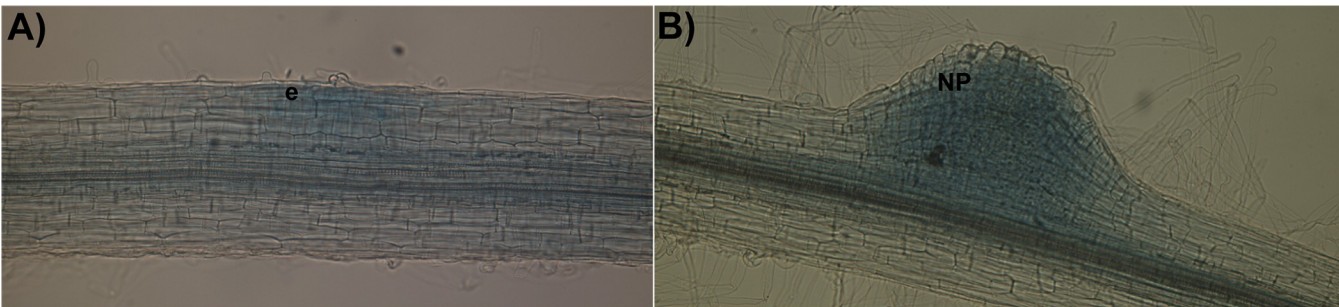

**Fig 2. Analysis of *RIPK2* promoter activity in transgenic hairy roots of common bean.** Promoter activity was determined based on GUS activity using X-Gluc as substrate. A) GUS activity in epidermal cells is restricted to one area of infection. B) GUS activity in a nodule primordium. e, epidermis; NP, nodule primordium.

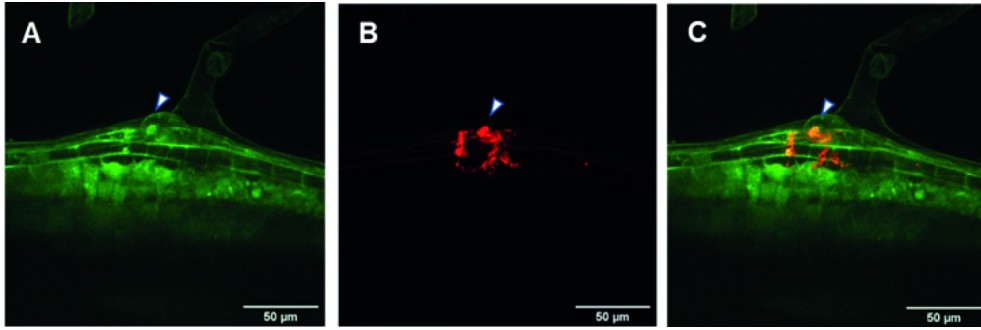

**Fig 3. Transgenic common bean hairy roots transformed with a plasmid expressing *PvAtg6-EGFP* and inoculated with *Rhizobium tropici* strain DS-red.** A) Green fluorescence signal (EGFP, 488-nm excitation and 510-nm emission) observed at the root infection site. B) Progression and branching of the IT based on the red fluorescence signal of *R. tropici* strain DS-red (543-nm excitation and 583-nm emission). C) Merged images of A) and B). Images were obtained *in vivo* under an IX81 Olympus FV1000 inverted confocal microscope (60× objective, S/1.3 NA). Arrows indicate the IT in transgenic roots, which were analyzed 8 days after rhizobial inoculation.

## Discussion

Composite plants are widely used to study the molecular mechanisms underlying plant developmental programs, particularly those regulating symbiosis between common bean roots and arbuscular mycorrhizal fungi or rhizobia [2, 3, 5]. Here, we present an update of the original

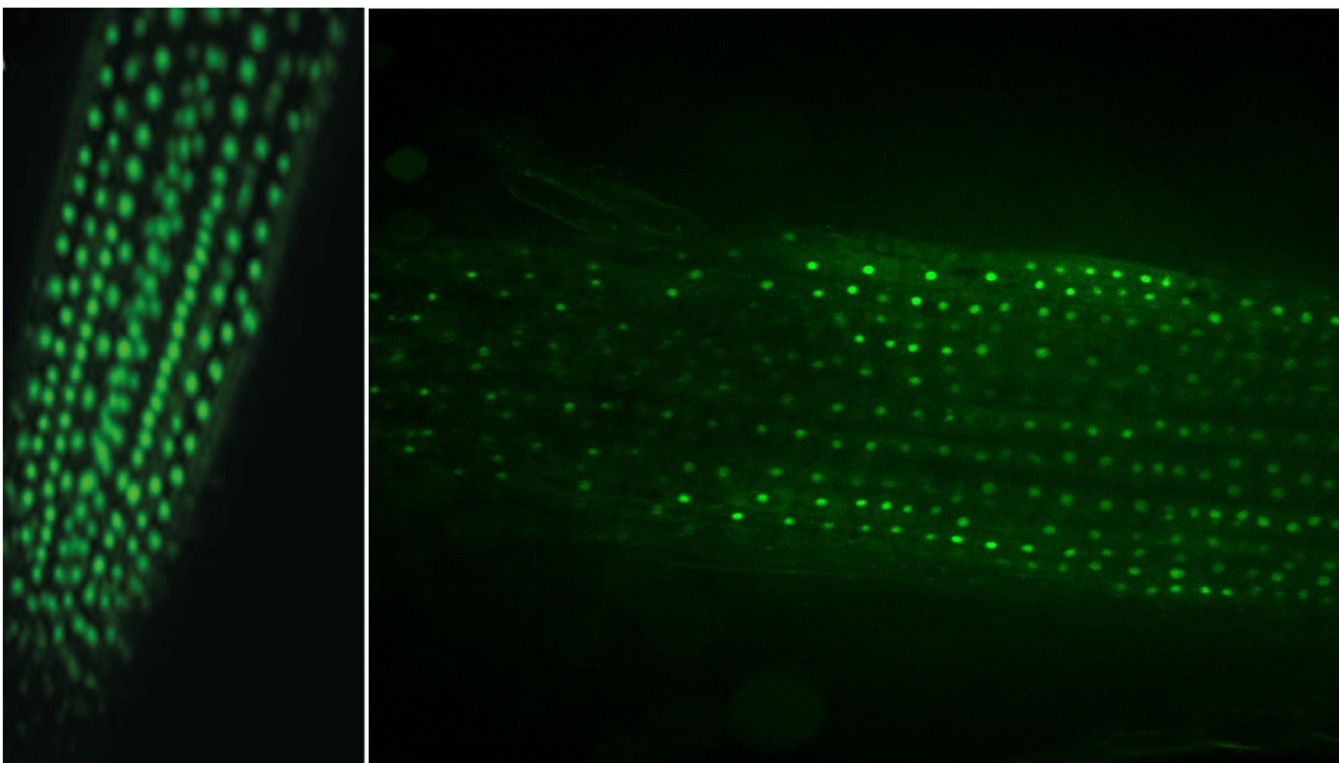

**Fig 4. Transgenic common bean hairy roots transformed with pH7FWG2Cas9.** The plasmid pH7FWG2Cas9 carries an open reading frame containing the *Cas9* coding sequence cloned in-frame and upstream of *EGFP* driven by the cauliflower mosaic virus (CaMV) 35S promoter. Green dots represent nuclear localization of Cas9, as revealed by observing green fluorescence (EGFP, 488-nm excitation and 510-nm emission) using a Nikon Eclipse Ti microscope in combination with a Yokogawa CSU-W1 spinning-disk confocal system.

protocol for generating transgenic hairy roots in common bean. Our results provide evidence that this updated protocol can be successfully used to study rhizobial symbiosis using reverse-genetics-based approaches. Using RT-qPCR analysis, we show that the expression of a given gene, in this case, *pPLA-IIγ*, significantly increased and the expression of *PvPLDα2* significantly decreased in transgenic common bean hairy roots (Fig 1) generated using our protocol. *pPLA-IIγ* and *PvPLDα2* roots were transformed with plasmid 35S:*pPLA-IIγ-EGFP* or *PvPLDα2*-RNAi, respectively. These results indicate that the updated protocol can be used to generate transgenic hairy roots with altered gene expression. Additionally, we successfully used this protocol to analyze the activity of promoters in specific cell types of transgenic hairy roots, as well as to track the progress of ITs (Figs 2 and 3); this strategy can be used to better understand the regulatory mechanisms in rhizobial symbiosis. Finally, our results suggest that this protocol could be used for gene editing based on the CRISPR/Cas9 system (Fig 4).

It is difficult to obtain stable knockout lines of common bean because this model plant is recalcitrant to transformation. Therefore, generating composite plants is the only available alternative for performing molecular analysis using reverse-genetics approaches. The main disadvantage of this method is the low percentage of plants that successfully develop transgenic hairy roots. In most cases, this low efficiency is due to a combination of the low intrinsic efficiency of *A. rhizogenes* clones in transforming seedling roots and suboptimal experimental practices during *A. rhizogenes*–mediated inoculation and the incubation of the inoculated plants.

Typically, good *A. rhizogenes* clones generate transgenic hairy roots within 10–12 days after inoculation. If a specific clone does not induce transgenic hairy roots after this incubation period, it should be discarded. If plants incubated with *A. rhizogenes* generate at least some transgenic hairy roots during the first 12 days after incubation, it is important to cut off the primary root (wild-type root) approximately 1 cm below the transgenic hairy root callus. The composite seedlings should then be transferred to glass tubes containing B & D medium [11], and the transgenic hairy roots should be maintained above the surface of the medium, keeping them inside the tubes for 3–5 days. These are critical steps for stimulating transgenic hairy root growth.

Because the wild-type (non-inoculated) root will later be excised, it is important to puncture the hypocotyls with a needle to create inoculation sites at a distance of ~1 cm from the radicle. This precaution will keep the transgenic hairy roots from growing too close to the wild-type roots, facilitating excision. Other factors that reduce the effectiveness of hairy root generation are excessive damage to hypocotyls with the tip of the needle and failure to remove the caps from the tubes 3–4 days after inoculation. In both cases, leaf growth will be severely damaged, which will hinder the generation of transgenic hairy roots.

To perform reverse-genetics analysis, plasmids introduced into *A. rhizogenes* for transformation should carry a visual selection marker gene, such as a gene encoding a fluorescent protein. Transgenic hairy roots should be selected based on the emission of fluorescence; therefore, *A. rhizogenes* clones transformed with a plasmid that produces non-fluorescent transgenic hairy roots should be discarded. Additionally, whether a plasmid overexpresses or downregulates the target gene must be validated by reverse-transcription quantitative PCR (RT-qPCR) before using the clones transformed with such plasmid for any experiments. This is important, as highly fluorescent transgenic hairy roots do not always show a high capacity for overexpression or silencing of the gene(s) under study. When generating transgenic hairy roots carrying an RNAi-based plasmid, it is advisable to select *A. rhizogenes* clones transformed with a plasmid able to induce silencing by at least 60% of transcript levels relative to non-transformed controls. Notably, gene silencing never fully abolishes transcript levels to 0% of that in the wild type. Conversely, for overexpression, we recommend selecting *A. rhizogenes* clones transformed with a plasmid able to increase the expression of the target gene to at least twice the level as that of the non-transformed controls. We highly recommend working with

two clones of *A. rhizogenes* transformed with the same plasmid with similar capacities for over-expression or silencing to guarantee reproducibility when analyzing phenotypes.

Another limitation of composite plants is that transgenes may themselves become silenced over time in transgenic hairy roots. Indeed, 3- to 4-week-old transgenic hairy roots can show a lower fluorescence intensity than younger transgenic hairy roots. Furthermore, the level of overexpression or silencing may vary among plants, which can affect reproducibility. Despite these disadvantages, since it is difficult to establish stable transgenic lines in common bean, generating composite plants is the preferred method for reverse-genetics studies of gene function. Importantly, we propose that producing transgenic hairy roots carrying a gene-editing construct based on the CRISPR/Cas9 system is a good strategy to efficiently knock out a gene.

## Conclusions

Composite plants of common bean are widely used to study the molecular mechanisms underlying symbiosis with arbuscular mycorrhizal fungi and rhizobia. However, the efficiency of hairy root generation in wild-type common bean plants is generally around 30%. Therefore, an inadequate and suboptimal procedure for generating transgenic hairy roots will yield only a few, if any, hairy root plants. This article, together with the previously published protocol (**https://dx.doi.org/10.17504/protocols.io.261ge3bpjl47/v2**), provides useful information and advice for improving the generation of transgenic hairy roots and their use in reverse-genetics studies. The number of hairy-rooted transgenic plants obtained depends on many factors, such as seed quality, plant growth conditions, and viability of the *A. rhizogenes* inoculum. Therefore, this document summarizes the changes and improvements made to the original protocol (S1 Table). The optimization and extension of the previous protocol are presented to be used successfully in reverse-genetics studies.

## Supporting information

**S1 Table. Comparison of the original published protocol for the generation of transgenic hairy roots with the updated protocol described here, considering the critical steps.** (DOCX)

**S1 File. Step-by-step protocol, also available on protocols.io (https://dx.doi.org/10.17504/protocols.io.261ge3bpjl47/v2).** (PDF)

**S2 File. Data set used for statistical analysis.** (DOCX)

## Acknowledgments

The authors are grateful to Xochitl Alvarado-Affantrange (Laboratorio Nacional de Microscopía Avanzada, UNAM) for assistance with confocal microscopy, Raul Muñoz Lezama (Universidad Autónoma del Estado de Morelos) for technical support in the greenhouse, and Olivia Santana (Instituto de Biotecnología, UNAM) for technical support in the preparation of *A. rhizogenes* strains. Moreover, we thank Unidad de Síntesis y Secuenciación at Instituto de Biotecnología, UNAM, for technical support with oligonucleotide synthesis and DNA sequencing.

## Author Contributions

**Conceptualization:** Ronal Pacheco, Georgina Estrada-Navarrete, Carmen Quinto.

**Data curation:** Ronal Pacheco, Georgina Estrada-Navarrete, Jorge Solis-Miranda, MA Juárez-Verdayes, Yolanda Ortega-Ortega.

**Formal analysis:** Ronal Pacheco, Carmen Quinto.

**Funding acquisition:** Carmen Quinto.

**Investigation:** Ronal Pacheco, Georgina Estrada-Navarrete, Carmen Quinto.

**Methodology:** Ronal Pacheco, Georgina Estrada-Navarrete, Noreide Nava.

**Project administration:** Carmen Quinto.

**Resources:** Carmen Quinto.

**Supervision:** Georgina Estrada-Navarrete, Carmen Quinto.

**Validation:** Carmen Quinto.

**Visualization:** Ronal Pacheco, Georgina Estrada-Navarrete, Carmen Quinto.

**Writing – original draft:** Ronal Pacheco.

**Writing – review & editing:** Ronal Pacheco, Carmen Quinto.

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
