## [Decision Letter · Decision Letter 0]

17 Jul 2023

PONE-D-23-18440Generation of common bean (Phaseolus vulgaris L.) composite plants and their proper use for reverse-genetics studiesPLOS ONE

Dear Prof. Carmen Quinto,

Thank you for submitting your manuscript to PLOS ONE. After careful consideration, we feel that it has merit but does not fully meet PLOS ONE’s publication criteria as it currently stands. Therefore, we invite you to submit a revised version of the manuscript that addresses the points raised during the review process.

We look forward to receiving your revised manuscript.

Kind regards,

Waqas Khan Kayani, PhD

Academic Editor

PLOS ONE

Journal Requirements:

Additional Editor Comments:

This manuscript presents some interesting new parameters for the induction of hairy roots. However, I believe it would be more beneficial (for researchers working in this field) if they could find a comprehensive comparison between the methods outlined in the previously published articles and the novel approach proposed in this manuscript. Instead of having to refer to both the previous publication and this one separately, a detailed comparison within this manuscript would be more advantageous. This way, readers can better understand the strengths and limitations of each method and make informed decisions when generating hairy roots from this plant.

Reviewers' comments:

Reviewer's Responses to Questions

**Comments to the Author**

1. Does the manuscript report a protocol which is of utility to the research community and adds value to the published literature?

Reviewer #1: Yes

Reviewer #2: Yes

2. Has the protocol been described in sufficient detail?

The step-by-step protocol should contain sufficient detail for another researcher to be able to reproduce all experiments and analyses.

Reviewer #1: Yes

Reviewer #2: Partly

3. Does the protocol describe a validated method?

Reviewer #1: Yes

Reviewer #2: Yes

4. If the manuscript contains new data, have the authors made this data fully available?

Reviewer #1: Yes

Reviewer #2: Yes

**5. Is the article presented in an intelligible fashion and written in standard English?**

Reviewer #1: Yes

Reviewer #2: Yes

6. Review Comments to the Author

Reviewer #1: The authors have described the modified method to produce composite Common bean plants with hairy roots. The transgenic approach for producing nutrient efficient plants is innovative approach for crop improvement. This protocol is good addition in the existing methods of producing transgenic plants. However the authors need to contextualize the discussion part of paper with existing references. As a whole the paper is fit for publication.

Reviewer #2: Reviewer Comments

The article is a lab protocol in which, nutritionally and economically a very important crop, common bean (Phaseolus vulgaris L.) is under study for development of hairy roots. An already published data of one of the authors in 2007, showed significantly efficient hairy root cultivars as mentioned in the manuscript. However, the current study is a modified protocol of the original protocol with genetic approaches included.

Title

The title of the manuscript should be more descriptive.

Abstract

The sentence making and grammar errors must kept in consideration.

Proper punctuation is required in the abstract to make effective writing.

No results are described in the abstract.

Introduction

Please correct the grammatical errors

Introduction is very well written, however, the significance of reverse genetics approaches to understand the molecular mechanisms regulating mycorrhizal and rhizobial symbioses should be incorporated.

Materials and Methods

Not enough data about the plasmid construct and genes is given.

Result

The result section should be the outcomes of the experiments performed despite of the expected results.

In the results section, there is no description of relative abundance for gene expression explained (Fig 1)..

There should be very explicit description of all the figures.

Discussion

Discussion section is very well explained, however, the literature insight on the results should be incorporated.

The discussion should have a strong link with the results explained

The overall manuscript is comprised of a comprehensive approaches related to the already published lab protocols on common bean hairy root culture. This data on composite plants development and use of gene editing tools (CRISPR Cas9) could give better understanding about the gene function in common bean plants.

7. PLOS authors have the option to publish the peer review history of their article (what does this mean?). If published, this will include your full peer review and any attached files.

Reviewer #1: **Yes: **Asad Hussain Shah

Reviewer #2: No

---

## [Author Response · Author response to Decision Letter 0]

29 Aug 2023

Reviewer Comments

The article is a lab protocol in which, nutritionally and economically a very important crop, common bean (Phaseolus vulgaris L.) is under study for development of hairy roots. An already published data of one of the authors in 2007, showed significantly efficient hairy root cultivars as mentioned in the manuscript. However, the current study is a modified protocol of the original protocol with reverse genetics approaches included.

R. We really appreciate the reviewer's comments, which helped us improve our manuscript.

Title

The title of the manuscript should be more descriptive.

R. We addressed your suggestion and modified the title as follows: “A comprehensive, improved protocol for generating common bean (Phaseolus vulgaris L.) transgenic hairy roots and their use in reverse-genetics studies”

Abstract

The sentence making and grammar errors must kept in consideration.

Proper punctuation is required in the abstract to make effective writing.

No results are described in the abstract.

R. We appreciate your feedback and have made the necessary changes to improve the wording in our manuscript. Since this report is not a classic article, but a protocol, it does not describe results as such, but "expected results." Therefore, a couple of sentences have been added that briefly describe the expected results. 

Introduction

Please correct the grammatical errors

Introduction is very well written, however, the significance of reverse genetics approaches to understand the molecular mechanisms regulating mycorrhizal and rhizobial symbioses should be incorporated.

R. We appreciate this comment. The Introduction was modified based on this suggestion.

Materials and Methods 

Not enough data about the plasmid construct and genes is given.

R. We agree with this comment. Information on plasmid construction has been added to the revised Materials and Methods section.

Result 

The result section should be the outcomes of the experiments performed despite of the expected results.

In the results section, there is no description of relative abundance for gene expression explained (Fig 1).

There should be very explicit description of all the figures.

R. We appreciate the suggestions. We modified the text accordingly.

Discussion

Discussion section is very well explained, however, the literature insight on the results should be incorporated.

The discussion should have a strong link with the results explained

R. We agree with these comments and modified the text in the Discussion based on the suggestions. 

The overall manuscript is comprised of a comprehensive approaches related to the already published lab protocols on common bean hairy root culture. This data on composite plants development and use of gene editing tools (CRISPR Cas9) could give better understanding about the gene function in common bean plants. 

R. We greatly appreciate the comments, as they helped us improve our manuscript.

---

## [Decision Letter · Decision Letter 1]

31 Oct 2023

A comprehensive improved protocol for generating common bean (Phaseolus vulgaris L.) transgenic hairy roots and their use in reverse-genetics studies

PONE-D-23-18440R1

Dear Dr. Carmen Quinto,

We’re pleased to inform you that your manuscript has been judged scientifically suitable for publication and will be formally accepted for publication once it meets all outstanding technical requirements.

Kind regards,

Waqas Khan Kayani, PhD

Academic Editor

PLOS ONE

Additional Editor Comments (optional):

Dear Dr. Carmen Quinto,

Gathering the reviewers' comments was a bit inconvenient, which resulted in a slight delay in processing this protocol. We hope this did not cause you much trouble.

Reviewers' comments:

Reviewer's Responses to Questions

**Comments to the Author**

1. Does the manuscript report a protocol which is of utility to the research community and adds value to the published literature?

Reviewer #2: Yes

2. Has the protocol been described in sufficient detail?

To answer this question, please click the link to protocols.io in the Materials and Methods section of the manuscript (if a link has been provided) or consult the step-by-step protocol in the Supporting Information files.

The step-by-step protocol should contain sufficient detail for another researcher to be able to reproduce all experiments and analyses.

Reviewer #2: Yes

3. Does the protocol describe a validated method?

Reviewer #2: Yes

4. If the manuscript contains new data, have the authors made this data fully available?

Reviewer #2: Yes

**5. Is the article presented in an intelligible fashion and written in standard English?**

Reviewer #2: Yes

6. Review Comments to the Author

Reviewer #2: The Author has made all the changes and corrections according to the recommendation in the manuscript

Overall manuscript is written in standard English language.

In introduction section literature related to the main theme is added.

Materials and methods section is very well explained.

Figures in the Results are well elaborated

Discussion is more comprehensive and relevant literature is added

7. PLOS authors have the option to publish the peer review history of their article (what does this mean?). If published, this will include your full peer review and any attached files.

Reviewer #2: **Yes: **Dr. Sammyia Jannat

---

## [Editor Report · Acceptance letter]

9 Nov 2023

PONE-D-23-18440R1 

A comprehensive, improved protocol for generating common bean (*Phaseolus vulgaris* L.) transgenic hairy roots and their use in reverse-genetics studies 

Dear Dr. Quinto:

I'm pleased to inform you that your manuscript has been deemed suitable for publication in PLOS ONE. Congratulations! Your manuscript is now with our production department. 

Kind regards, 

on behalf of

Dr. Waqas Khan Kayani 

Academic Editor

PLOS ONE